# Neurodevelopmental Outcome at 6 Months Following Neonatal Resuscitation in Rural Tanzania

**DOI:** 10.3390/children10060957

**Published:** 2023-05-27

**Authors:** Ingrid Ask Torvik, Robert Moshiro, Hege Ersdal, Anita Yeconia, Raphael Mduma, Jeffrey Perlman, Jørgen Linde

**Affiliations:** 1Faculty of Health Sciences, Stavanger University, 4036 Stavanger, Norway; hege.ersdal@safer.net; 2Stavanger University Hospital, 4068 Stavanger, Norway; jorgen.linde@safer.net; 3Department of Paediatrics, Muhimbili National Hospital, Dar es Salaam 65000, Tanzania; robert.moshiro@mnh.or.tz; 4Research Center, Haydom Lutheran Hospital, Haydom P.O. Box 9000, Tanzania; anita.yeconia@haydom.co.tz (A.Y.); raphaelmduma@gmail.com (R.M.); 5Department of Pediatrics, Weill Cornell Medical College, New York, NY 10065, USA; jmp2007@med.cornell.edu

**Keywords:** neonatal resuscitation, bag-mask ventilation, helping babies breathe, follow up, neurodevelopment

## Abstract

Early bag-mask ventilation (BMV) administered to non-breathing neonates at birth in the presence of birth asphyxia (interruption of placental blood flow) has reduced neonatal mortality by up to 50% in low- and middle-income countries. The neurodevelopmental outcome of neonates receiving BMV remains unknown. Using the Malawi Developmental Assessment Tool (MDAT), infants who received BMV at birth were assessed at 6 months, evaluating gross motor, fine motor, language and social skills. A healthy cohort with no birth complications was assessed with the same tool for comparison. Mean age-adjusted MDAT z-scores were not significantly different between the groups. The number of children having developmental delay defined as a z-score ≤ −2 was significantly higher in the resuscitated cohort for the fine motor and language domain and overall MDAT z-score. The prevalence of clinical seizures post discharge was significantly higher in the resuscitated group and was associated with neurodevelopmental delay. Infants with developmental delay or seizures were more likely to have a 5 min Apgar < 7 and a longer duration of BMV. Most children receiving BMV at birth are developing normally at 6 months. Still, there are some children with impaired development among resuscitated children, representing a subgroup of children who may have suffered more severe asphyxia.

## 1. Introduction

Intrapartum-related hypoxic events related to disruption of placental blood flow (birth asphyxia) are one of the leading causes of neonatal mortality and morbidity together with preterm delivery and infection [1,2,3]. Most of the mortality and morbidity occurs in sub-Saharan Africa and South Asia, yet there are limited studies that have evaluated the neurodevelopmental outcomes of such infants in the low-resource setting (LRS) [4]. There is a major difference between high-, middle- and low-income countries in the availability, accessibility and quality of health care facilities in maternal and neonatal care.

The World Health Organization (WHO) has broadly defined birth asphyxia as “failing to initiate or maintain regular breathing at birth” [5]. It is estimated that 10% of all newborns need some help at birth, with stimulation and/or clearing of the airways required for the infant to start breathing [6]. Additionally, 5–6% require positive pressure ventilation and are likely to be in secondary apnea [7] due to prolonged interruption of fetal–placental blood flow [6,8].

Over the past decade, there has been an increased focus on resuscitation at birth, with the implementation of the Helping Babies Breathe (HBB) program, an evidence-based neonatal resuscitation education curriculum for LRSs, now implemented in more than 80 countries [9]. The algorithm includes when to start stimulation, suction and positive pressure ventilation, while heart compression or epinephrine is not part of the HBB guidelines. Resuscitation training of HBB in low- and middle-income countries has been shown to decrease mortality by 20 to 50% [10,11].

Survivors of birth asphyxia are at risk for neurodevelopmental disabilities, including motor deficits, learning and neuropsychological difficulties, autism, epilepsy, hearing and sensory loss [12,13]. It has been estimated that 1.15 million babies develop neonatal encephalopathy associated with intrapartum events, with 96% born in low-income countries [14]. In high- and middle-income regions with improved survival rates, increasing disability has been noted [1]. With improved survival after resuscitation in low- and middle-income countries, it has become increasingly important to address and determine the long-term development and the resulting burden on the health system and society.

Evaluating neurodevelopmental outcomes is challenging. In a rural LRS, following children over time is especially difficult due to lack of infrastructure, people moving frequently and high transport cost. Neurodevelopment is a complex process affected by multiple factors such as genetic, social, economic and environmental factors, nutrition and health status. Poverty, poor health and nutrition are estimated to be responsible for more than 200 million children under 5 years of age failing to reach their developmental potential [15].

There are few existing studies on developmental outcomes following birth asphyxia in a rural low-resource setting after implementing basic resuscitation. The BRAIN-HIT study, a multicenter study in India, Pakistan and Zambia found normal development at 3 years of age following neonatal resuscitation [16]. Amongst survivors of neonatal encephalopathy (NE) in Uganda, 29.3% had neurodevelopmental impairments, and 40% had a completely normal outcome at 27 months [17]. The infants with an abnormal outcome had already developed neurologic signs in the neonatal period. There is a need for more knowledge on the developmental outcomes of all resuscitated children in LRSs. Early outcomes have not previously been studied and are of particular interest to detect delays that might benefit from early interventions. The aim of this study was to evaluate early neurodevelopmental outcomes at 6 months of age in infants who received BMV administered due to being non-breathing at birth compared to a healthy cohort in rural Tanzania.

## 2. Materials and Methods

### 2.1. Setting

Haydom Lutheran Hospital is situated in the Manyara region in north-east Tanzania, 300 km from the nearest urban city Arusha. The hospital has 900.000 people in its catchment area, and 5.7 million people in its larger referral area [18]. There are approximately 3500–4000 deliveries each year. The HBB curriculum and training program was implemented in 2009 and led to the establishment of the Safer Births project in 2013 [19]. The Safer Births register gathered detailed information on deliveries and neonatal resuscitation, providing a unique opportunity to follow children and identify perinatal risk factors for adverse outcomes. Neonatal resuscitation is usually led by midwives following the HBB guidelines [9]. If needed, newborns are admitted to the neonatal care unit (NCU), which offers antibiotics, continuous positive airway pressure (CPAP), intravenous fluids, phenobarbitone for clinical seizures and nutrition but not invasive or mechanical ventilation, therapeutic hypothermia, electroencephalogram (EEG) or brain imaging. We did not have reliable data on neonatal seizures and hypoxic ischemic encephalopathy (HIE) status from the hospital nor were the infants examined for other congenital anomalies.

Subjects lived in a vast agricultural area around the hospital. Children living within a 20 km radius of the hospital were included for follow up. Locating children was challenging due to poor infrastructure, high transport costs, not all families having a phone and participants moving from one area to another. Families were visited 2–3 times to be located and asked for consent and then the infants were assessed as close to 6 months of age as possible.

### 2.2. Study Design

This was a prospective cohort study of neurodevelopmental outcomes at six months of age in children who received bag-mask ventilation at Haydom Hospital from January 2020 to August 2021 compared to a cohort of children who had no complications at birth. Resuscitated children were recruited at the hospital after delivery, and names, phone numbers and home locations were stored in the study files. Inclusion criteria for the resuscitated cohort were: 1. received bag-mask ventilation at birth; 2. gestational age ≥ 35 weeks; 3. birth weight > 2500 g; 4. no history of traumatic brain injury.

For each resuscitated child, a child of the same gender and age from the same village or region was located and included to account for potential endemic factors and other setting-related factors in the village. Inclusion criteria for the healthy cohort were: 1. normal birth without any complications; 2. gestational age ≥ 35 weeks; 3. not admitted to hospital in the first month of life; 4. no history of traumatic brain injury. For the comparison cohort, inclusion criteria were confirmed by parents as the infants were recruited from their homes not through the hospital or birth records.

All infants were examined as close to six months as possible, giving ±1 month for the resuscitated cohort and ±3 months for the comparison cohort. Due to COVID-19, data collection was paused for 6 weeks, and some participants lost their target window. A local cognitive team of 7 research nurses visited the families at their homes, assessing neurodevelopmental scoring and filling out a household questionnaire mapping socioeconomic aspects of the family, child health status and measured child growth. Parents were asked if the child had seizures, defined as the whole or part of the body shaking or loss of consciousness. If yes, they were asked if the infant used any seizure medication. No further investigation for diagnosing epilepsy was made for this study, though infants using seizure medication were assumed to have been diagnosed at a pediatric clinic. Research nurses were not blinded to the two cohorts, as they also were responsible for recruiting controls when visiting villages.

Figure 1a,b illustrates the inclusion process. To check for selection bias, we compared 1 and 5 min Apgar scores and time to ventilation and duration of ventilation between those who completed the MDAT (n = 159) and those not located, those that refused and those that died (n = 66).

### 2.3. Neurodevelopmental Scoring

The Malawi Developmental Assessment Tool (MDAT) is a culturally relevant tool proven to be reliable and validated for detecting neurodevelopmental disabilities in children from 0 to 6 years [20]. It possesses four domains: gross motor, fine motor, language (both receptive and expressive language skills) and social skills. A normal reference range defines which items to include at different ages. Assessment starts for an item within the normal reference range for each domain. The child is tested until they fail 6 consecutive items, then is tested backwards from the starting point until they pass 6 consecutive items and then automatically passes all items below this. Each item passed gives 1 point, resulting in a summarized score for each domain and a total MDAT score. All research nurses received three days MDAT training from a certified trainer. We used an adapted version of the MDAT, which was already adapted, piloted and validated for Haydom in previous studies [21]. Testing was performed in Swahili or the local language Iraqw. Language and social domain adaptions were performed for words or questions that did not directly translate to Swahili. Gross motor and fine motor domains did not require adaption. All results were transformed into age-adjusted z-scores using the MDAT scoring app, making scores comparable to the original cohort of 1426 children in Malawi [22].

Assessments were videotaped for quality control purposes. The cognitive team reviewed videos frequently to assure agreement of scoring. After the study period, some videos were additionally reviewed by a pediatrician (JL) and neurologist (IT). Videos of children with a very low or high score, and videos of children that seemed to have an unlikely scoring (such as failing easy items and passing more advanced items), were selected for this second review. Forty videos were reviewed in depth (23 resuscitated, 17 controls). Eight resuscitated and one healthy infant had minor changes applied to their scoring due to video scoring (Appendix A). Videos confirmed a high quality and consistent performance of scoring.

### 2.4. Statistical Analysis

All data were double-entered by data clerks using EpiData 3.1 (EpiData Association, Odense, Denmark). Further data processing and analyses were performed using SPSS (IBM SPSS Statistics for Windows, version 26.0; IBM Corp., Armonk, NY, USA). Child growth measurements were transformed to age-adjusted z-scores using WHO SPSS syntax [23,24]. To compare results between the groups, we used the Pearson chi-square or Fisher’s exact test for categorical variables and the independent sample *t*-test or Mann–Whitney U test for continuous variables, as appropriate. We also performed ad hoc analyses comparing the number of children with a z-score ≤ −2 and ≤−1 for each domain, reflecting children with severe or more subtle developmental delays. Further subgroup analysis was performed looking at 1 and 5 min Apgar scores, duration of ventilation and time from birth to start of ventilation in the resuscitated infants with developmental delays compared to those resuscitated with a normal outcome (z-score > −2).

### 2.5. Statistical Power

We estimated the recruitment of 175 resuscitated newborns and 175 non-resuscitated newborns based on a power calculation with a stipulated mean score difference of 30% in the Malawi Developmental Assessment Tool given an alpha level of 0.05 and beta level of 0.80 [25].

### 2.6. Ethical Considerations

The study was approved by the Tanzanian National Institute of Medical Research (NIMR) number NIMR/HQ/R.8a/Vol.IX/3036 and the Norwegian Regional Committee for Medical and Health Research Ethics (REK) 2018/2407/REK. Data handling was general data protection regulation (GDPR) compatible. Parents were asked for written consent during follow up. In case of illiteracy, an impartial witness was present during consenting and documented his/her name, and the parent signed with a thumbprint. The parents of children with any MDAT warning signs or reported seizures were advised to seek health care services, and an appointment with a pediatrician at Haydom Lutheran Hospital was facilitated when needed or upon request from the parents.

## 3. Results

### 3.1. Baseline Characteristics

In total, 159 resuscitated infants who received BMV (Figure 1a) and 175 comparable healthy controls (Figure 1b) were included. There were no difference in Apgar score < 7 at 1 min, 5 min and time from birth to start of ventilation and duration of ventilations among the resuscitated who were lost to follow up (n = 66) compared to those included in this study (n = 159), Appendix A.

There were no significant differences between the groups related to parental education, housing tenure, gender, twin status, nutritional factors and vaccination status. All infants were vaccinated according to the Tanzanian immunization program. Table 1 shows characteristics of socioeconomic status and health conditions between the two cohorts at follow up. In the resuscitated cohort, more parents reported having a previous stillbirth and owning a car and/or a bicycle. There was a significant difference in the infant’s mean age at the time of assessment, i.e., resuscitated infant mean days 188.5 (SD 7.2, 95% CI 187.35–189.60), healthy cohort 192.6 (SD 13.4, 95% CI 190.58–194.60) (*p* = 0.001). All further analyses were age adjusted. Mother’s mean age at birth and number of siblings were significantly higher in the healthy cohort (*p* = 0.037 and 0.001, respectively). There was no significant difference in number of family members living together. More families in the healthy cohort responded that all eligible children in the family attended school (73% and 56%, respectively, *p*-value = 0.001). 

Table 2 presents infant growth measurements. There were no significant differences in length for age (stunting), weight for age, weight for length (wasting) and head circumference between the groups. Stunting, defined as length for age ≤ −2, was found in 19% of children in both groups.

### 3.2. Neurodevelopmental Outcomes

Mean z-scores for gross motor (GM), fine motor (FM), language (LA), social skills (SO) and total MDAT for the resuscitated cohort were GM 0.005, FM 0.554, LA 0.006, SO 0.713 and MDAT 0.103 and, for the healthy cohort, GM 0.123, FM 0.801, LA −0.052, SO 0.744 and MDAT 0.258 (*p*-values GM 0.130, FM 0.084, LA 0.643, SO 0.708, MDAT 0.174), see Table 3.

Figure 2a–c illustrates the distribution of scores. (All domain figures can be found in Appendix A.) The majority in both groups were within the normal range, but there was a tail of extreme outliers with a poor outcome in the resuscitated group.

Infants with a z-score ≤ −2 were defined as having a neurodevelopmental impairment. Of the resuscitated cohort, 7.5% compared to 2.9% of the non-resuscitated cohort had an impairment in one or more domains (*p* = 0.051), see Table 4.

For the fine motor, language and total MDAT scores, the difference between the groups was significant (*p*-values: FM 0.041, LA 0.024, MDAT 0.030). The same analyses were performed with the cut-off of a z-score ≤ −1, which could indicate more subtle neurodevelopmental delays. There were no differences between the groups in any domain (*p*-values GM 0.092, FM 0.060, LA 0.168, SO 0.202, MDAT 0.071), Table 5.

Amongst resuscitated infants, seven (4.4%) were reported to have had clinical seizures at the 6-month follow up compared to one (0.6%) in the comparison cohort (*p* = 0.023). This also accounted for six out of seven infants in the resuscitated cohort with an overall MDAT score ≤ −2 (ranging from −2.3 to −6.5). In the resuscitated cohort, an antiepileptic drug (phenobarbitone) was used in one infant; additionally, four children had used phenobarbitone for 1–2 months. One of these infants was seizure free since and had a normal development with an MDAT z-score of 0.04. From the healthy cohort, the infant reporting seizures was using phenobarbitone, was reported to being seizure free and had a z-score of −0.8.

In the resuscitated cohort, we performed subgroup analysis looking at Apgar score at 1 and 5 min, time from birth to ventilation and duration of ventilation in children with any z-score < −2 and/or reporting seizures. This revealed that infants with any z-score ≤ −2 or reporting seizures at 6 months were more likely to have an Apgar score at 1 and 5 min < 7 and longer duration of BMV (*p*-values 0.025, < 0.001 and 0.001, respectively). Time from birth to start of ventilation did not differ between the groups (Table 6). Head circumference for age z-scores at 6 months were significantly lower in the resuscitated group with adverse outcome compared to the resuscitated group with a normal outcome (mean −1.55, 95% CI −2.36–−0.74, SD 1.27 vs. 0.17 (95% CI 0.00–0.33, SD 1.02, *p* < 0.001)).

## 4. Discussion

Birth asphyxia, reflecting the inability to initiate respirations at birth, increases the risk of neonatal mortality and neurodevelopmental impairment in survivors. Timely and effective bag-mask ventilation reduces neonatal mortality. Few studies have investigated neurodevelopment following resuscitation in the LRS [16,17,26,27]. In this study, we found no significant mean differences in the overall MDAT or specific domains between the groups. This is in accordance with findings from the BRAIN-HIT study [16]. However, we found an increased prevalence of infants with clinical seizures and delays in the fine motor and language domains amongst the resuscitated infants. In addition, there was a relation between clinical seizures and developmental delay.

In contrast to other studies, which reported up to 29% of survivors of HIE/NE who exhibited NDI [17,26], this study focused on all infants in need of BMV at birth. Though mean scores did not differ, we identified a subgroup of children with NDI defined as any domain z-score ≤ −2. In the resuscitated cohort, there were significantly more children with delay in the fine motor and language domains and lower total MDAT. In three of the six infants failing the language domain, the parent reported that the child did not respond to sound, indicating hearing impairment. Numbers were few, with only 12 (7.5%) in the resuscitated and 5 (2.9%) in the healthy cohort with any NDI. Other studies reported 35% of birth asphyxia survivors in LRSs exhibiting any NDI [27]. Our numbers could underestimate impairment as both groups had positive mean scores, i.e., they might be overscored. Additionally, the cut-off of ≤ −2 might not identify minor impairments. Still, analysis of z-scores ≤ −1 did not differ between the groups. These positive findings might illustrate that given timely and effective treatment, survivors of birth asphyxia may have an outcome comparable to other children in this rural low-resource setting.

The majority, 92.5%, of resuscitated infants had development within the normal range. Amongst these, there could be children with minor delays and impairments not detectable at this early age. Indeed, some impairments may only become evident in later childhood [28], and some may have improved development and catch up with their potential given optimal health and social conditions. Still, it is important to detect delays early to help children reach their full potential. The BRAIN-HIT study in India, Pakistan and Zambia showed a trend of poorer scores at 12 months but a greater increase in developmental scores at 36 months among resuscitated compared to non-resuscitated [16] infants and found early, home-based intervention to be beneficial and cost effective [29,30].

Parental reports of seizures were significantly higher in the resuscitated group compared to the non-resuscitated group. Though numbers were few, there seemed to be a clear overrepresentation of seizures in the resuscitated group compared to the estimated incidence of early onset epilepsy (0–59 months), which is estimated at 61.7/100.000 children per year [31]. However, in Uganda, 5.7% reported childhood seizures in the comparison cohort. Our numbers could be underreported in both cohorts as we lacked EEG data and accurate seizure description. Further, there was a strong relation between seizures and adverse developmental outcomes, consistent with previous reports [17]. Frequent seizures might enhance poor development, or there might be a common pathway of brain injury causing seizures and poor development.

Further analysis of perinatal factors amongst the resuscitated infants with any NDD or seizure compared to those with a normal outcome showed that these were more likely to have a 5 min Apgar < 7 and a longer duration of PPV, suggestive of more severe asphyxia. They also had significantly lower head circumference at 6 months. Whether this was related to the hypoxia, malnutrition or congenital disorders was not identified in this study.

Socioeconomic factors and child growth measurements were generally comparable between the groups, though there was a high male-to-female ratio in both groups. In the resuscitated cohort, more parents reported owning a car and/or bicycle and had experienced previous stillbirths. This may be selection bias, as they had a motivation and opportunity to seek health care facilities for delivery. We did not have information about whether previous stillbirths were fresh or macerated. Fresh still births might be related to placentation issues, also posing a risk for a depressed newborn. The healthy cohort had a higher mean maternal age and parity, as was also found in other studies [16,17]. This might be related to primiparity and young maternal age possessing a risk for birth asphyxia [32]. There was no significant difference in number of family members (including all people living together) who could also play a part in stimulating and interacting with the child.

### Strengths and Limitations

Strengths: This was one of the largest studies yet to follow resuscitated newborns in a low-resource rural setting, where the issue of birth asphyxia (disruption of placental blood flow) is most relevant. Examiners used a standardized and validated scoring system (MDAT), and assessments were video-recorded. We had a healthy cohort for comparing the development of children in the same surroundings and produced z-scores to make scores more comparable to other studies.

Limitations included the short-term follow up, as six months is too early to detect developmental impairments such as cerebral palsy and learning difficulties, and delayed children might catch up and achieve their full potential later in childhood. The cognitive team was not blinded as they were also responsible for locating and recruiting controls, possibly causing bias. During the study, two nurses quit, and two new nurses started in June 2021, resulting in a total of seven different nurses assessing the children. To increase the inter-rater reliability, study nurses visited families in pairs and reviewed the videos frequently to assure agreement of scoring. The comparison group did not have data from birth, and was included based on parental recall. Diagnosis of epilepsy was not verified; seizures and treatment were based on parental report. The study was designed to find a mean score difference of 30% between the groups, and the sample size might not have been large enough to identify smaller differences.

## 5. Conclusions

This is the largest study to date that has evaluated short-term outcomes in infants requiring resuscitation since the implementation of HBB in many low-resource settings. It adds new information on the early neurodevelopment of resuscitated children in a rural LRS. No significant overall differences between resuscitated and non-resuscitated children in mean MDAT scores at 6 months were found. However, the prevalence of impairments was significantly higher for those who received BMV. Importantly, approximately 93% of resuscitated children had neurodevelopmental scores within the normal range at six months. The resuscitated children were a heterogeneous group with different degrees of birth asphyxia, and it seems that a poor outcome at this age is related to the severity of asphyxia. There is still a great need for longer-term follow up of resuscitated newborns in order to identify those who need early interventions.

## Figures and Tables

**Figure 1 children-10-00957-f001:**
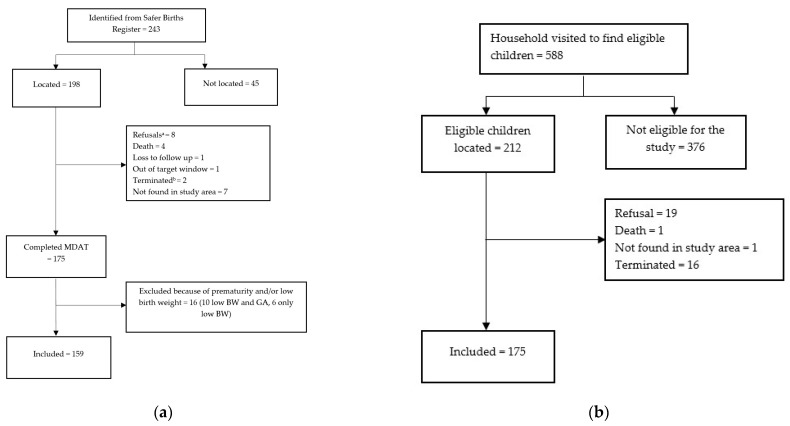
(**a**) Resuscitated cohort, (**b**) healthy cohort. ^a^ Refused to participate in study, ^b^ terminated: Gender or date of birth did not fit with the research database, or we were not able to assess them within the target window.

**Figure 2 children-10-00957-f002:**
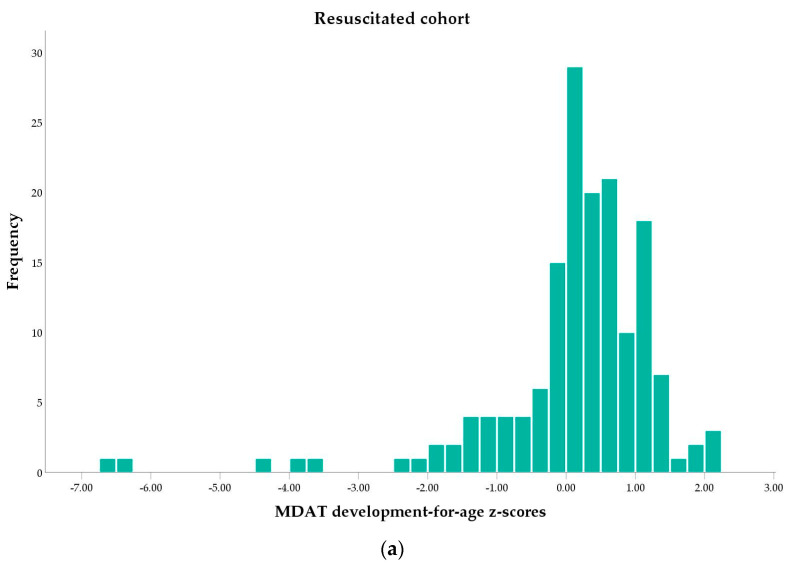
Neurodevelopmental outcomes as the distribution of age-adjusted z-scores of the Malawi Developmental Assessment Tool (MDAT) in the resuscitated cohort (**a**) and healthy cohort (**b**). The boxplot (**c**) demonstrates extreme outliers in the resuscitated cohort. Circles (ο) represent potential outliers within 1.5–3 interquartile ranges (IQR) from the quartile 1. Asterix (*) represents extreme outliers that are more than 3 IQR from the quartile 1.

**Table 1 children-10-00957-t001:** Baseline characteristics and socioeconomic factors.

	Resuscitated, n = 159	Healthy Controls, n = 175 *	*p*-Value **
**Gender**			
Female Male	65 (40.9 %) 94 (59.1 %)	75 (42.9 %) 100 (57.1 %)	0.715
**Child’s age at assessment in days** (Mean, SD, min–max)	188.5 (7.2, 187.35–189.60)	192.6 (13.4, 190.58–194.60)	**0.001**
**Twins**			
First born Second born Singleton	2 (1.3%) 3 (1.9%) 154 (96.9%)	6 (3.4%) 1 (0.6%) 168 (96.0%)	0.259
**Mothers age at delivery** (mean, SD, 95% CI)	26.5 (7.2, 25.39–27.64) ^c^	28.2 (SD 6.5, 27.19–29.14) ^c^	**0.037**
**Position amongst siblings**	2.86 (2.43, 2.86–3.23)	3.75 (2.23, 3.4–4.1)	**<0.001**
**Exclusive breastfeeding**	151 (95.0%)	171 (98.3%)	0.092
**Months of exclusive breastfeeding** (mean, SD, 95% CI)	4.47 (1.65, 4.21–4.73)	4.85 (1.24, 4.61–4.98)	0.246
Number of months:			0.394
0 months 1 months 2 months 3 months 4 months 5 months 6 months	8 (5.1%) 3 (1.9%) 9 (5.7%) 16 (10.1%) 26 (16.5%) 43 (27.2%) 53 (33.5%)	3 (1.7) 1 (0.6%) 5 (2.9) 18 (10.3%) 30 (17.2%) 57 (32.8%) 60 (34.5%)	
**Marital status**			
Married Single	138 (86.8%) 21 (13.2%)	158 (90.8%) 16 (9.2%)	0.245
**Primary caretaker education**			
No formal education Primary school Secondary school College and above	14 (8.8%) 95 (59.7%) 38 (23.9%) 12 (7.5)	14 (8.0%) 110 (63.2%) 42 (24.1%) 8 (4.6%)	0.700
**Paternal education**	(n = 156)	(n = 171)	0.243
No formal education Primary school Secondary school College and above	13 (8.3%) 91 (58.3%) 35 (22.4%) 17 (10.9%)	14 (8.2%) 117 (68.4%) 27 (15.8%) 13 (7.6%)	
**Previous stillbirth**	16 (10.1%)	7 (4.0 %)	**0.031**
**Housing**			
Concrete house with floor and ceiling Concrete house without floor or ceiling Traditional housing	25 (15.7) 80 (50.3) 54 (34.0)	31 (17.8) 95 (54.6) 48 (27.6)	0.447
**Number of family members** Mean (SD, 95% CI)	7.28 (3.42, 6.8–7.8)	7.01 (2.7, 6.6–7.4)	0.886
**Crowding index ^a^** Mean (SD, 95% CI)	2.53 (1.41, 2.31–2.75)	2.66 (1.14, 2.49–2.83)	**0.036**
**Items of possession**			
Radio TV Mobile phone Smart phone Bicycle Car Tractor Refrigerator Business Owns none of the above	61 (38.4) 22 (13.8) 148 (93.1) 36 (22.6) 101 (63.5) 10 (6.3) 2 (1.3) 4 (2.5) 21 (13.2) 6 (3.8)	58 (33.1) 23 (13.1) 161 (92.0) 37 (21.1) 90 (51.4) 1 (0.6) 1 (0.6) 1 (0.6) 32 (18.3) 9 (5.1)	0.320 0.853 0.708 0.741 **0.026** **0.003** 0.607 0.196 0.205 0.546
**All children in the family attend school**	89 (56.0%)	127 (73%)	**0.001**
**Vaccination**	159 (100%)	174 (100%)	
**Malnutrition reported by parents**	0 (0%)	1 (0.6%)	1.00
**Feeding problem ^b^**	1 (0.6%) ^a^	0 (0%)	0.477
**Started to eat solid food at same age as other children**	128 (81%)	135 (78%)	0.503
**How often has the child been sick**			
Very often Once a month Once every two months Once in 6 months Very rare	10 (6.3) 0 2 (1.3%) 2 (1.3%) 145 (91.2%)	6 (3.4%) 4 (2.3%) 0 0 164 (94.3%)	**0.029**
**Needed help from health care professionals ^c^**	1 (0.6%)	0 (0%)	0.296

* Missing data: one missing piece of information from household questionnaire. In addition, missing data relating to eating solid food (1 resuscitated, 1 healthy), exclusive breastfeeding (1 resuscitated), needed help from health care personnel (1 healthy), any child born dead (1 healthy), paternal education (3 resuscitated, 3 healthy). ** *p*-values with chi-square or Fisher’s exact test, as appropriate. *t*-test or Mann–Whitney U test for continuous variables, as appropriate. ^a^ Crowding index = number of people per room living in the house. ^b^ Feeding problem: was not able to suck. ^c^ Needed help from health care professional: hospital doctor due to convulsions.

**Table 2 children-10-00957-t002:** Child growth measurements.

Growth Measurement	Resuscitated, n = 159	Healthy, n = 174	*p*-Value
Length for age z-score	−0.97 (1.17, −1.15–−0.79)	−1.06 (1.16, −1.23–−0.89)	0.449
Weight for age z-score mean	0.08 (1.07, −0.09–0.24)	−0.07 (1.23, −0.26–0.11)	0.204
Weight for length z-score	0.98 (1.01, 0.82–1.14)	0.85 (1.17, 0.67–1.03)	0.242
Head circumference for age z-score	0.05 (1.12, −0.12–0.23)	−0.07 (0.94, −2109–0.07)	0.334
Ad hoc analysis of z-score ≤ −2:			
Length for age	30 (18.9%)	33 (19.0%)	0.982
Weight for age	4 (2.5%)	10 (5.7%)	0.142
Weight for length	2 (1.3%)	2 (1.1%)	1.00
Head circumference for age	7 (4.4%)	3 (1.7%)	0.203

**Table 3 children-10-00957-t003:** Neurodevelopmental outcomes at 6 months presented as age-adjusted mean scores for each domain and total MDAT.

	Resuscitated, n = 159	Healthy, n = 175	*p*-Value
	Mean (SD *)	Mean (SD *)	
Gross motor	0.005 (0.84)	0.123 (0.55)	0.130
Fine motor	0.554 (1.5)	0.801 (1.0)	0.084
Language	0.006 (1.38)	−0.052 (0.83)	0.643
Social	0.713 (0.79)	0.744 (0.71)	0.708
MDAT *	0.103 (1.25)	0.258 (0.75)	0.174

* Abbreviations: MDAT: Malawi Developmental Assessment Tool, SD: standard deviation.

**Table 4 children-10-00957-t004:** Neurodevelopmental delays.

Age-Adjusted Z-Score ≤ −2	Resuscitated, n = 159	Healthy, n = 175	*p*-Value
Gross motor	2 (1.3%)	0	0.226
Fine motor	11 (6.9%)	4 (2.3%)	**0.041**
Language	5 (3.1%)	0	**0.024**
Social	2 (1.3%)	1 (0.6%)	0.607
MDAT	7 (4.4%)	1 (0.6%)	**0.030**
Any domain *	12 (7.5%)	5 (2.9%)	0.051
Reported seizures at 6 months	7 (4.4%)	(n = 174) 1 (0.6%)	**0.030**
Any domain or report of seizures	13 (8.2%)	6 (3.4%)	0.061

* Some children had a poor score in several domains, reflected as the number of any z-score less then minus 2 being lower than the sum of numbers above.

**Table 5 children-10-00957-t005:** Number of children with z-scores ≤ −1.

Age-Adjusted Z-Score ≤ −1	Resuscitated, n = 159	Healthy, n = 175	*p*-Value
Gross motor	7 (4.4%)	2 (1.1%)	0.092
Fine motor	23 (14.5%)	14 (8.0%)	0.060
Language	18 (11.3%)	29 (16.6%)	0.168
Social	7 (4.4%)	3 (1.7%)	0.202
MDAT	19 (11.9%)	11 (6.3%)	0.071
Any domain	36 (22.6%)	46 (26.3%)	0.440

**Table 6 children-10-00957-t006:** Neonatal factors and outcome.

	Any MDAT ≤ −2 or Seizure	Normal Outcome	*p*-Value
Apgar 1 min 7–10	3 (23.1%)	81 (55.5%)	
Apgar 1 min < 7	10 (76.9%)	65 (44.5%)	0.025
Apgar 5 min 7–10	6 (46.2%)	132 (90.4%)	
Apgar 5 min < 7	7 (53.8%)	14 (9.6%)	<0.001
Time to first ventilation (s) Median, interquartile range	72 (35)	86 (77)	0.206
Duration of ventilation (s) Median, interquartile range	270 (471)	90 (123)	0.001

## Data Availability

Datasets may be available upon reasonable request to the corresponding author.

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
