# Peer review of "Neurodevelopmental Outcome at 6 Months Following Neonatal Resuscitation in Rural Tanzania"

_children, 2023, doi:10.3390/children10060957_

Round 1
Reviewer 1 Report
The severity of asphyxia has an important effect on the neurological prognosis of infants. It would have been better to pay attention to the severity of asphyxia in infants
Author Response
Dear reviewer
Thank you for reviewing and approving this article. I agree that the severity of asphyxia deserves more attention. However, the aim of this study was to describe the outcome of the whole group of infants receiving positive pressure ventilation at birth, as this has been little studied before and represents a large population of children globally. We are currently working on an other study looking more specific at severity of asphyxia and predictors of outcome in the same studypopulation.
Kind regards
Ingrid Ask Torvik
Reviewer 2 Report
Dear authors,
This is a very important study on developmental outcomes following birth asphyxia in a rural low-resource setting, after implementing basic resuscitation. I only have some minor comments and recommendations.
-Line 97-98 - please define EEG and HIE
- The supplementary material is not available
- Were the seizures correlated with other pathology (eg infections, fever)?
- Were the newborns resuscitated at birth also evaluated for the presence of associated anomalies (for example cardiac or pulmonary anomalies)?
Author Response
Dear reviewer
Thank you for reviewing this article, and constructive feedback.
I've resovled your suggestion and questions as followed;
- We spelled out electroencephalogram (EEG) and hypoxic ischemic encephalopathy (HIE) (line 97-98)
- Please find the supplementary material attached
- Seizures were not specified if correlated to infection or other health issue. As this only was a small part of a wide questionnaire it would be to extensive to map the seizures further. However, if reporting seizure the parents were asked if the child used antiseizure drugs (AED). If using AED, this strongly suggest that the child has been seeing a paediatrician giving the diagnose of epilepsy. As most of the children did use or had been using AED, they are likely to have visited a paediatrician finding indication for AED, and not beeing related to fever or other sickness. Some clarification has been added in line 127-130 in the updated article; “If answering yes they where asked if the infant used seizure mediaction. No further investigation for diagnosing epilepsy where made for this study, though infants using seizure medication are assumed to have been diagnosed at a paediatric clinic.”
- No, the newborns resuscitated at birth were not evaluated for the presence of associated anomalies (for example cardiac or pulmonary anomalies). In line 99 in the updated article this has been added for clarification; "Nor where the infantes examined for other congenital anomalies.”
Best regards
Ingrid Ask Torvik
